# Taming Modality Entanglement in Continual Audio-Visual Segmentation

**Yuyang Hong**[1,2]   **Qi Yang**[1,2]   **Tao Zhang**[2,1]
**Zili Wang**[1,2]   **Zhaojin Fu**[3]   **Kun Ding**[2]   **Bin Fan**[3]   **Shiming Xiang**[1,2,*]
[1]**School of Artificial Intelligence, UCAS,** [2]**MAIS, Institute of Automation, CAS,**
[3]**School of Intelligent Science and Technology, University of Science and Technolog Beijing**
`{hongyuyang2023,yangqi2021,zhangtao2021}@ia.ac.cn`
* **Corresponding author**

Reviewed on OpenReview: `https://openreview.net/forum?id=8mPymf31zG`

## Abstract

Recently, significant progress has been made in multi-modal continual learning, aiming to learn new tasks sequentially in multi-modal settings while preserving performance on previously learned ones. However, existing methods mainly focus on coarse-grained tasks, with limitations in addressing modality entanglement in fine-grained continual learning settings. To bridge this gap, we introduce a novel Continual Audio-Visual Segmentation (CAVS) task, aiming to continuously segment new classes guided by audio. Through comprehensive analysis, two critical challenges are identified: 1) multi-modal semantic drift, where a sounding objects is labeled as background in sequential tasks; 2) co-occurrence confusion, where frequent co-occurring classes tend to be confused. In this work, a Collision-based Multi-modal Rehearsal (CMR) framework is designed to address these challenges. Specifically, for multi-modal semantic drift, a Multi-modal Sample Selection (MSS) strategy is proposed to select samples with high modal consistency for rehearsal. Meanwhile, for co-occurence confusion, a Collision-based Sample Rehearsal (CSR) mechanism is designed, allowing for the increase of rehearsal sample frequency of those confusable classes during training process. Moreover, we construct three audio-visual incremental scenarios to verify effectiveness of our method. Comprehensive experiments demonstrate that our method significantly outperforms single-modal continual learning methods. The source code will be made publicly at `https://github.com/cqu-student/CAVS-CMR`.

## 1 Introduction

Humans are inherently capable of continuously learning while retaining knowledge from previous tasks. For example, infants can progressively recognize new animals while remembering those they have already learned. This human ability has motivated extensive research into continual learning Wang et al. (2024), which enables models to learn sequential tasks. Early work Rebuffi et al. (2017); Bang et al. (2021); Sun et al. (2023) primarily focused on classification, employing techniques such as regularization or rehearsal to mitigate catastrophic forgetting. Subsequent methods Cermelli et al. (2020) have extended continual learning to semantic segmentation. However, when directly applied to multi-modal (e.g. audio-visual) scenarios, these single-modal methods exhibit suboptimal performance Mo et al. (2023).

Recently, several methods Mo et al. (2023); Pian et al. (2023; 2025); Yang et al. (2023) have extended continual learning to multi-modal scenarios. For example, AV-CIL Pian et al. (2023) proposes a continual audio-visual classification method with a dual similarity constraint enforcing both instance-level and class-level cross-modal semantic consistency. ContAV-Sep Pian et al. (2025) proposes a framework for audio-visual separation that incorporates cross-modal similarity distillation to preserve semantic consistency between modalities. Meanwhile, real-world applications require fine-grained audio-visual continual learning. For

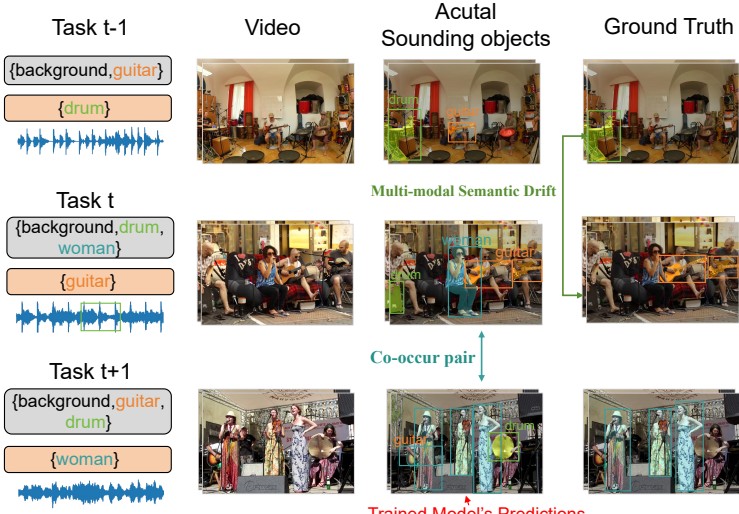

Figure 1: Illustration of CAVS and two challenges. In the figure, three sequential tasks are presented from top to bottom. Gray boxes: learned or background classes, light-orange boxes: target classes to be learned. Multi-modal semantic drift occurs when a learned class (e.g., darkgreen drum) is labeled as background in task $t$, despite the presence of its corresponding sound in the audio. This drift causes the model to suffer catastrophic forgetting of the modality semantic associations specific to the drum. Co-occurrence confusion occurs when, in a previous task (e.g. task $t$), two classes frequently co-occur (guitar and woman). After learning a new task, the model tends to misclassify the old classes (guitar) as the new ones (woman).

example, embodied intelligence needs to identify the source of a vocalization from environmental audio-visual cues. However, existing methods primarily focus on coarse-grained audio-visual tasks and therefore fail to address fine-grained tasks, such as disentangling pixel-level visual features from audio signals under continual learning scenarios.

Meanwhile, recent research Zhou et al. (2022; 2024); Yang et al. (2024) has explored fine-grained modality entanglement between audio signals and visual features in audio-visual segmentation. AVSBench Zhou et al. (2022) establishes the first benchmark for aligning the pixel-level visual semantics with the corresponding audio signals. COMBO Yang et al. (2024) further explores bilateral relations of three entanglements, pixel, modality, and temporal, to enhance the model's representational capacity. However, audio-visual segmentation cannot be directly applied to continual learning scenarios, as it is designed for static settings.

To this end, we introduce a novel fine-grained multi-modal continual learning task, termed **C**ontinual **A**udio-**V**isual **S**egmentation (**CAVS**). Specifically, CAVS needs to perform audio-visual segmentation in a sequential task setting while retaining knowledge of previously seen classes. To address CAVS, we reformulate the AVS Zhou et al. (2022; 2024) framework and adapt classical continual semantic segmentation methods to the audio-visual context. Based on our observations, we identify two new challenges in fine-grained continual learning tasks: (1) Multi-modal semantic drift: Incorrect audio-visual semantic alignment (e.g. drum-background) due to mislabeling of learned classes as background exacerbates catastrophic forgetting. (2) Co-occurrence confusion: Frequent co-occurrence of categories leads to modality entanglement, for example, the audio modality of woman becomes entangled with visual modality of guitar in Fig. 1. In essence, these two issues are manifestations of modality entanglement from different perspectives.

To tackle these challenges, we propose a Collision-based Multi-modal Rehearsal (CMR) framework. Specifically, a collision is the discrepancy between the predictions and the ground truth labels during rehearsal. To the best of our knowledge, this is the first rehearsal-based framework specifically designed for the audio-visual continual scenario. For challenge (1), Multi-modal Sample Selection (MSS) is introduced, which leverages additional single-modal models to select multi-modal samples with high modal consistency for rehearsal, thereby enhancing inter-modal alignment (correct audio-visual entanglement). For challenge (2), Collision-

based Sample Rehearsal (CSR) is proposed, which dynamically adjusts the class ratio of samples for rehearsal based on the collision frequency between the old model's predictions and the ground truth labels. In this process, classes with higher collision frequencies (defined as the discrepancy between the predictions and the ground-truth labels) are identified as classes that are more prone to be confused with newly learned classes. By increasing the number of rehearsal samples from classes with high collision frequency, the model can better leverage the audio modality to distinguish confusing classes, thereby mitigating catastrophic forgetting during training.

To validate the effectiveness of CMR, we reformulate the audio-visual dataset AVSBench Zhou et al. (2022) into three sequential task setup to better simulate a continual learning scenario. Specifically, our datasets include (1) AVSBench-Class Incremental (AVSBench-CI), (2) AVSBench-Class Incremental for Single-object (AVSBench-CIS), and (3) AVSBench-Class Incremental for Multi-object (AVSBench-CIM). Comprehensive experiments demonstrate that our proposed method achieves encouraging performance, showcasing its ability to effectively address the multi-modal semantic drift and co-occurrence confusion in CAVS.

Our main contributions can be summarized as follows:

- We pioneer the extension of continual learning to audio-visual segmentation, introducing the Continual Audio-Visual Segmentation (CAVS). To the best of our knowledge, this is the first work to address audio-visual segmentation in a continual learning setting.

- For multi-modal semantic drift, we propose a Multi-modal Sample Selection (MSS) strategy to identify high-quality multi-modal samples with enhanced modal consistency. To solve co-occur confusion, we introduce a Collision-based Sample Rehearsal (CSR) mechanism where the rehearsal frequency of learned classes is dynamically adjusted based on collision frequency.

- Extensive experiments on three class-incremental datasets demonstrate that our method achieves state-of-the-art performance, validating its effectiveness in continual audio-visual segmentation.

## 2 Related Works

### 2.1 Continual Learning.

Continual learning focuses on incrementally training models to adapt to new tasks while preserving knowledge from previously learned ones. Recently, many works Rebuffi et al. (2017); Bang et al. (2021); Li et al. (2024); Kang et al. (2022); Ostapenko et al. (2019) have proposed regularization-based and rehearsal-based methods to address the problem of catastrophic forgetting. Rehearsal-based methods Rebuffi et al. (2017); Bang et al. (2021); Sun et al. (2023); Li et al. (2024); Kang et al. (2022) allow for the storage of a small subset of old data in memory, which is later utilized for rehearsal during training. iCaRL Rebuffi et al. (2017) introduces a strategy to identify and retain the most representative samples for each class, which are replayed during training to mitigate forgetting in class-incremental learning. Pseudo-sample rehearsal-based methods Ostapenko et al. (2019); Shin et al. (2017); Wu et al. (2018) utilize generative models to create pseudo-samples of old classes. DGR Shin et al. (2017) establishes an initial framework where learning each new task is coupled with replaying the data generated by the old generative model. Building upon continual learning Rebuffi et al. (2017); Li et al. (2024); Kang et al. (2022); Ostapenko et al. (2019); Shin et al. (2017); Wu et al. (2018), Class-Incremental Semantic Segmentation (CISS) requires pixel-level classification to achieve fine-grained segmentation Cermelli et al. (2020); Douillard et al. (2021); Zhang et al. (2022); Yang et al. (2023); Cha et al. (2021); Yin et al. (2025). PLOP Douillard et al. (2021) suggests generating pseudo-labels by identifying latent past classes within the current background. ScaleSeg Yang et al. (2023) employs prototypes refined through online contrastive clustering and incorporates a background diversity strategy to boost plasticity. While Cermelli et al. (2020) addressed semantic shifts within a single modality, our work reveals more complex multi-modal semantic drift where modal consistency is considered.

## 2.2 Multi-modal Continual Learning

Recent works have extended continual learning to multi-modal scenarios. AV-CIL Pian et al. (2023) proposes a continual audio-visual classification method with a dual similarity constraint enforcing both instance-level and class-level cross-modal semantic consistency. ContAV-Sep Pian et al. (2025) introduces a framework for audio-visual separation that incorporates cross-modal similarity distillation to preserve semantic consistency between modalities. CIGN Mo et al. (2023) proposes a class-incremental grouping network that leverages audio-visual grouping to support continual learning of new categories. However, these methods primarily focus on coarse-grained tasks such as classification or sound separation, and do not address the fine-grained pixel-level segmentation challenges that arise in CAVS. Our work is the first to extend continual learning to audio-visual segmentation, where modality entanglement at the pixel level introduces unique challenges not present in classification settings.

## 2.3 Audio Visual Segmentation

Audio-visual segmentation (AVS) is a novel and challenging task that localizes sound sources in visual scenes by pixel-level prediction Zhou et al. (2022; 2024); Yang et al. (2024). AVSBench Zhou et al. (2022) establishes the first audio-visual segmentation benchmark and introduces the Temporal Pixel-wise Audio-Visual Interaction (TPAVI) module to incorporate audio semantics as guidance for visual segmentation. AVSeg-Former Zhou et al. (2024) develops a transformer-based framework with audio queries, learnable queries, and an audio-visual mixer for selective attention and dynamic feature adjustment. CATR Li et al. (2023) proposes a combinatorial fusion framework that captures audio-visual spatiotemporal dependencies through cross-modal interaction modelling. ECMVAE Mao et al. (2023) decomposes audio and visual data in latent space, explicitly modeling both shared and modality-specific representations to enhance segmentation performance. COMBO Yang et al. (2024) rethinks AVS by exploring the bilateral relations of three entanglements, pixel, modality, and temporal, to enhance the model's representation ability. In this work, we develop a framework for continual learning scenarios, making the task more aligned with real-world applications.

## 3 Methods

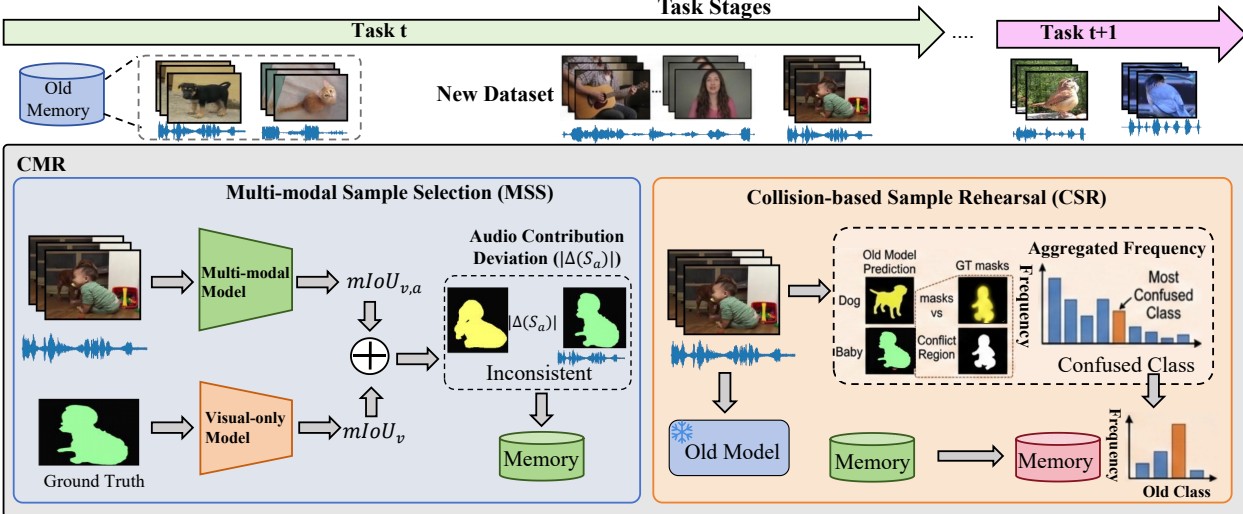

Figure 2: Overview of the proposed CMR framework. The CMR framework introduces a novel rehearsal-based method for continual audio-visual segmentation. Our method consists of two key modules: (a) Multi-modal Sample Selection (MSS) strategy for samples rehearsal, which identifies samples with high modality consistency by computing the difference in mean Intersection-over-Union ($mIoU$) between uni-modal and multi-modal models. (b) Collision-based Sample Rehearsal (CSR) strategy that dynamically adjusts the rehearsal frequency of samples based on the collision between the old model and current ground truth.

The proposed CMR framework, as illustrated in Fig 2, is constructed based on the ResNet50 architecture from AVSBench. The subsequent sections first revisit continual semantic segmentation, followed by a formal formulation of CAVS. Subsequently, we present the two core components of our framework: multi-modal sample selection and collision-based sample rehearsal.

For clarity, we summarize the key notations used throughout this paper in Tab. 1.

Table 1: Summary of key notations.

| Symbol | Description |
|---|---|
| $\mathcal{C}^t$ | Set of new classes introduced at learning stage $t$ |
| $\mathcal{Y}^t$ | Cumulative label space up to stage $t$: $\mathcal{Y}^t = \bigcup_{i=0}^{t} \mathcal{C}^i$ |
| $\mathcal{D}_t$ | Training dataset at stage $t$ |
| $S = (\{S_v^k\}, S_a)$ | A sample with $T$ video frames and one audio signal |
| $f_{\theta^t}^{v,a}$ | Audio-visual segmentation model at stage $t$ |
| $f_{\theta^t}^{v}$ | Visual-only segmentation model at stage $t$ |
| $\Delta(S_a)$ | Audio modality contribution: $mIoU_{v,a} - mIoU_v$ |
| $M_t$ | Memory buffer constructed at stage $t$ |
| $\mathcal{F}_c$ | Collision frequency of old class $c$ |
| $\mathcal{T}$ | Threshold for collision ratio filtering |
| $k$ | Number of samples selected per class for memory |

## 3.1 Revisiting Continual Semantic Segmentation (CSS)

CSS assumes that tasks arrive sequentially, with each task containing a set of categories $\mathcal{C}^t$ and a corresponding training set $\mathcal{D}_t$, where $t$ denotes the current learning stage.

The goal of the learning task $\mathcal{D}_t$ at a given stage $t$ is to learn a model $f_\theta^t$ parameterized by $\theta^t$ to accurately predict the label given an input image $X$. The predicted output segmentation mask for pixel $i$ can be computed as: $\max \{f_{\theta^t}(X)[i, c]\}^{|\mathcal{Y}^t|} - 1_{c=0}$, where $f_{\theta^t}(X)[i, c]$ denotes predicted probability of class $c$ at pixel $i$.

In this setting, CSS assumes that tasks arrive sequentially, with each task $\mathcal{D}_t$ containing a set of categories $\mathcal{C}^t$ that are disjoint from those in other tasks. Training occurs in multiple phases, referred to as learning steps, where data from previous tasks may not be accessible in subsequent steps. Specifically, CSS further assumes that the previous $t - 1$ tasks encompass categories $\mathcal{Y}^{t-1} = \bigcup_{i=0}^{t-1} \mathcal{C}^i$, and task $\mathcal{D}_t$ introduces new categories $\mathcal{C}^t$. The model $f_{\theta_t}$ trained on the current task $\mathcal{D}_t$, while leveraging the previous model $f_{\theta_{t-1}}$ and avoiding catastrophic forgetting. In this work, we extend this setting to continual audio-visual segmentation.

## 3.2 Problem Setup and Notation of CAVS

For CAVS, the input space is defined as $\mathcal{S} \subset \mathcal{X} \times \mathcal{A}$, where $\mathcal{X}$ and $\mathcal{A}$ represent the visual and audio modalities, respectively. Each input sample $S = (\{S_v^k\}, S_a) \in \mathcal{D}_t$ contains $T$ consecutive video frames paired with an audio signal $S_a$, where $T = 10$. We formally define the following object categories: **Sounding objects** refer to objects in the visual scene that correspond to the active audio signal (e.g., a guitar being played). **Non-sounding objects** are visible objects that do not correspond to any active sound in the current audio. **Previously learned sounding objects** (old classes) are classes learned in prior stages $\mathcal{C}^{1:t-1}$, for which the model must retain segmentation capability without access to their original training data. The sounding objects in the $k$-th video frame $S_v^k$ are annotated with pixel-level labels. The objective of the $t$-th learning stage is to learn a model $f_{\theta^t}^{v,a} : \mathcal{S} \mapsto \mathbb{R}^{N \times |\mathcal{C}^t|}$, where $N$ is the number of pixels per frame. The audio-visual segmentation mask for pixel $i$ in frame $S_v^k$ can be computed as:

$$\hat{y}_i^k = \arg \max_{c \in \mathcal{Y}^t} f_{\theta^t}^{v,a}(\{S_v^k\}, S_a)[i, c]. \tag{1}$$

In contrast, for task $\mathcal{D}_t$, both non-sounding objects from $\mathcal{Y}^t$ and sounding objects from $\mathcal{Y}^{t-1}$ are assigned the background label, while the audio $S_a$ remains unchanged. That is, at each stage $t$, the label space

expands from $\mathcal{Y}^{t-1}$ to $\mathcal{Y}^t = \mathcal{Y}^{t-1} \cup \mathcal{C}^t$. Samples from previous stages stored in the memory buffer retain their original annotations; new training data is labeled only for $\mathcal{C}^t \cup \{background\}$, with pixels of old classes treated as background in the new data, following the standard class-incremental segmentation protocol. Compared to AV-ICL Pian et al. (2023), CAVS demands more substantial fine-grained alignment between global audio cues and local visual semantics. The CAVS formulation directly generalizes the well-established class-incremental segmentation (CIS) paradigm to the audio-visual domain. The disjoint/overlapped settings and the background labeling strategy follow standard CIS conventions Cermelli et al. (2020); Douillard et al. (2021), ensuring that the protocol is not benchmark-specific but reflects real-world deployment scenarios where an agent encounters new sound-producing object categories over time.

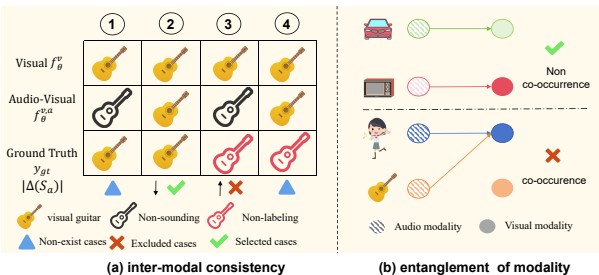

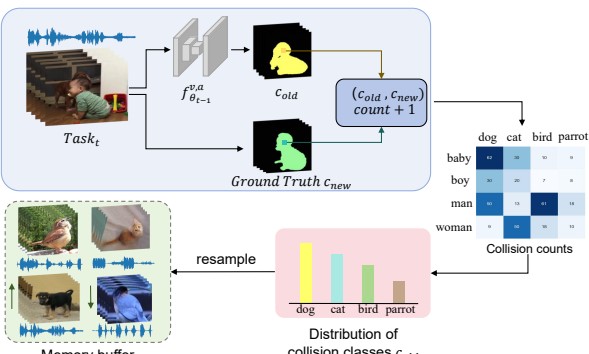

Figure 3: Illustration of inter-modal consistent samples and entanglement of modality. **(a)** Cases 1 and 4 don't appear in practice because selection uses already well-trained samples where audio and video predictions match the ground truth. Case 3 represents samples characterized by multi-modal semantic drift and is typically excluded due to substantially large $|\Delta(S_a)|$. Conversely, Case 2 is kept because of its cross-modal semantic consistency. **(b)** Classes with infrequent co-occurrence exhibit weak audio-visual entanglement, while frequent co-occurrence leads to strong cross-modal entanglement (e.g., guitar sounds and images of women).

Figure 4: Illustration of the collision-based sample rehearsal: for each sample, we calculate conflicts between old model predictions (dog) and current ground truth (baby). Aggregating these across all samples yields the collision frequency $\mathcal{F}$, quantifying confusion between old and new classes. By aligning the distribution of replayed samples with the collision frequency, the model is better guided to disentangle incorrect modality semantic associations during training.

## 3.3 Multi-modal Sample Selection

Multi-modal semantic drift occurs when learned classes are mislabeled as background in new tasks, which in turn leads to the incorrect modality semantic associations. Therefore, replaying samples with consistent modality semantics helps alleviate the multi-modal semantic drift of previously learned classes in the current task. However, as shown in Fig. 1, existing selection strategies fail to identify samples with high modality semantic consistency and may instead select samples that contain multi-modal semantic drift.

Inspired by the work in Wei et al. (2024a), where Shapley values are leveraged to quantify uni-modal contributions to model predictions, we propose a Multi-modal Sample Selection (MSS) strategy. By quantifying the contribution of the audio modality, this strategy identifies samples with high inter-modal consistency for rehearsal. Formally, given a video sample $S = (\{S_v^k\}, S_a) \in \mathcal{D}_t$, we train two parallel models:

$$f_{\theta_t}^v(\{S_v^k\}) : \mathcal{X} \mapsto \mathbb{R}^{N \times |\mathcal{Y}^t|}, \tag{2}$$

$$f_{\theta_t}^{v,a}(\{S_v^k\}, S_a) : \mathcal{S} \mapsto \mathbb{R}^{N \times |\mathcal{Y}^t|}. \tag{3}$$

After training, we compute the $mIoU$ scores for both modalities: visual-only model performance $mIoU_v$ and audio-visual model performance $mIoU_{v,a}$.

$$mIoU_v = \mathcal{J}_{mean}(f_\theta^v(\{S_v^k\}), \{y_{gt}^k\}), \tag{4}$$

$$mIoU_{v,a} = \mathcal{J}_{mean}(f_\theta^{v,a}(\{S_v^k\}, S_a), \{y_{gt}^k\}), \tag{5}$$

As illustrated in Fig. 3 (a), samples exhibiting smaller $\Delta(S_a)$ exhibit reduced multi-modal semantic drift. Therefore, $\Delta(S_a)$ is used to select samples that are more suitable for rehearsal. Calculation of $\Delta(S_a)$ is as follows:

$$\Delta(S_a) = mIoU_{v,a} - mIoU_v, \tag{6}$$

where $y_{gt}$ is the ground truth of video frame $S$, $\mathcal{J}_{mean}$ denotes computation of averaged $mIoU$ over $T$ frames.

For each newly added class $c \in \mathcal{C}^t$, we select the top-$k$ samples with the smallest absolute audio contribution deviation $|\Delta(S_a)|$ from $\mathcal{D}_t$ to construct the memory buffer $M_t$. Here, $\Delta(S_a)$ is computed *per sample $S$*, not per class. The top-$k$ (default $k = 5$) refers to selecting the $k$ samples *for each class $c \in \mathcal{C}^t$* with the smallest $|\Delta(S_a)|$, not the number of frames. We require consistency over all $T$ frames because audio-visual alignment is a temporal property, a sample with inconsistent per-frame performance suggests unstable modality correspondence, which is undesirable for rehearsal.

These selected samples are stored and replayed during the training of subsequent tasks through $\mathcal{D}_{t+1} \cup M_t$, which effectively reinforces cross-modal associations. Our ablation studies demonstrate that this criterion outperforms random selection by 2.0 mIoU (see Tab. 3), highlighting the importance of modality consistency in sample rehearsal.

It is worth noting that a small $|\Delta(S_a)|$ does not imply that the audio modality contributes minimally to the prediction, nor does it indicate visual dominance. Instead, it reflects that the audio and visual modalities are consistently aligned—i.e., the audio signal is coherent with the visual content. Prior work Li et al. (2026) has shown that AVS models can exhibit visual bias, segmenting objects based on visual cues alone. In contrast, MSS explicitly selects samples where cross-modal semantics are stable: both modalities agree on the segmentation output. This is fundamentally different from visual bias, as we are selecting samples where audio-visual alignment is reliable, rather than where audio is irrelevant.

### 3.4 Collision-based Sample Rehearsal

As shown in Fig. 3 (b), frequently co-occurring classes in the old task will exhibit incorrect modality entanglement because of confusion in the audio modality.

To be more specific, frequent occurrence pulls the two classes closer in the feature space, which causes confusion. By aligning the distribution of replayed samples with the collision frequency, we increase the rehearsal frequency of collision classes, thereby promoting the disentanglement of incorrect modality semantic associations.

To implement this idea, we propose the Collision-based Sample Rehearsal (CSR) strategy, which identifies classes prone to co-occurrence confusion by detecting collisions between the old model's predictions and the ground truth. As illustrated in Fig. 4, for a new sample $S$, a collision occurs when the old model $f_{\theta_{t-1}}^{v,a}$ predicts an old class $c_{old} \in \mathcal{Y}^{t-1}$ in a spatial position where the ground truth $c_{new} \in \mathcal{C}^t$ appears.

Specifically, with the old model and task $\mathcal{D}_t$, the collisions between the prediction of $f_{\theta_{t-1}}^{v,a}$ and $\mathcal{D}_t$ is first computed. Inferring the video $S$ with the old model $f_{\theta_{t-1}}^{v,a}$, we obtain a collision pair

---

**Algorithm 1** Collision-Based Sampling

**Require:** Old model $f_{\theta^{t-1}}^{v,a}$, Training dataset $\mathcal{D}_t$, Semantic label $\mathcal{Y}_{gt}$, Threshold $\mathcal{T}$

**Ensure:** Collision frequency $\mathcal{F}$

1: **for** $S_i \in \mathcal{D}_t$ **do**
2:     Compute $\hat{\mathcal{Y}} \leftarrow f_{\theta^{t-1}}^{v,a}(S_i)$
3:     Mask $\mathcal{M} \leftarrow (\hat{\mathcal{Y}} \neq \text{background}) \wedge (\mathcal{Y} \neq \text{background})$
4:     Collision Region $\mathcal{I} \leftarrow (\hat{\mathcal{Y}} \neq \mathcal{Y}) \wedge \mathcal{M}$
    ▷ Count Pairs:
5:     **for** $i \in \mathcal{I}$ **do**
6:         $\text{Collision}(\hat{\mathcal{Y}}_i, \mathcal{Y}_i) \leftarrow \text{Collision}(\hat{\mathcal{Y}}_i, \mathcal{Y}_i) + 1$
7:     **end for**
    ▷ Get Most Confused Class:
8:     $(\mathcal{C}_{\text{old}}, \mathcal{C}_{\text{new}}) \leftarrow \arg\max \text{Collision}(\hat{\mathcal{Y}}_\mathcal{I}, \mathcal{Y}_\mathcal{I})$
9:     $\mathcal{R} \leftarrow \dfrac{\text{Collision}(\mathcal{C}_{\text{old}}, \mathcal{C}_{\text{new}})}{\sum \text{Collision}(\hat{\mathcal{Y}}_\mathcal{I}, \mathcal{Y}_\mathcal{I})}$
    ▷ Update Frequency:
10:    **if** $\mathcal{R} > \mathcal{T}$ **then**
11:        $\mathcal{F}_{\mathcal{C}_{\text{old}}} \leftarrow \mathcal{F}_{\mathcal{C}_{\text{old}}} + 1$
12:    **end if**
13: **end for**
14: **return** $\mathcal{F}$

---

$(c_{old}, c_{new})$. Since the old model has not trained
on new samples, it can only predict old classes
$c_{old} \subset \mathcal{Y}^{t-1}$. Assuming that the predicted result is $c_{old}$ and the ground truth label is $c_{gt}$, we count all collision pairs $(c_{old}, c_{new})$ and identify the learned class with the highest number of collisions as the most confusing class for the current video $S$:

$$\mathcal{P}(S) = \arg\max\{Count(c_i, c_j)|i \in \mathcal{Y}^{t-1}, j \in \mathcal{C}^t\}. \tag{7}$$

Next, the ratio $R$ of the number of collisions for the most confusing old class to the total number of collisions in a single frame $S$ is calculated as:

$$R_c = \frac{Count(\mathcal{P}(S) = c)}{\sum\{Count(c_i, c_j)|i \in \mathcal{Y}^{t-1}, j \in \mathcal{C}^t\}}, \tag{8}$$

$R_c$ denotes the ratio of $c$. if $R_c$ is greater than $\mathcal{T}$, which is the mean ratio across all learned classes, then we record that this old class has caused a significant collision. Note that $\mathcal{T}$ is not a manually tuned hyperparameter but is adaptively computed as the mean collision ratio at each incremental stage. This design ensures that CSR automatically adjusts to the current task distribution without requiring threshold tuning. This process will be repeated for all samples to obtain the collision frequency $\mathcal{F}$ of learned class:

$$\mathcal{F}_c = \sum_{i=1}^{D_t}(P(S_i) = c) \wedge (R_c > \mathcal{T}), \tag{9}$$

where *Count* represents the current number of collisions, and $\mathcal{F}_c$ indicates the collision frequency for class c in the current dataset $\mathcal{D}_t$.

The collision frequency for classes that do not exhibit collisions will be set to 1. To prevent the collision frequency of certain classes from becoming excessively large, we apply sigmoid smoothing. The results are then normalized to obtain $\mathcal{F}'$, as in Eq. equation 10.

$$\mathcal{F}' = \frac{sigmoid(\mathcal{F})}{\sum sigmoid(\mathcal{F})}. \tag{10}$$

Specifically, classes without collisions have $\mathcal{F}_c = 1$ (mapping to sigmoid$(1) \approx 0.731$), while classes with collisions have $\mathcal{F}_c > 1$ (mapping to values in $(0.731, 1.0)$). The purpose of the sigmoid function is to *compress* the distribution, preventing a single highly-colliding class from dominating the resampling budget. After normalization, the result is a valid probability distribution where collision-heavy classes receive proportionally more rehearsal samples, but not excessively so.

With $\mathcal{F}'$, 20% of the original memory $M_{t-1}$ is first sampled and then combined with the existing memory $M_{t-1}$, resulting in $\hat{M}_{t-1}$. In $\hat{M}_{t-1}$, samples from easily confused classes account for a larger proportion. Replaying $\hat{M}_{t-1}$, the model can more effectively distinguish between confusable classes, thereby mitigating the problem of catastrophic forgetting.

To provide a more comprehensive elaboration on Collision-based Sample Rehearsal, we provide its algorithmic procedure in Alg. 1. The algorithm demonstrates how we leverage collisions to identify categories affected by co-occurrence-induced semantic confusion, and further quantifies their replay frequency by tracking how often such misclassifications occur across inference samples.

The Multi-modal Sample Selection and Collision-based Sample Rehearsal methods effectively reduce forgetting on classes prone to multi-modal semantic drift and co-occurrence confusion, enhancing the model's capability for CAVS. The experiments demonstrate that rehearsal with resampling yields superior performance compared to direct rehearsal.

## 4 Experiments

### 4.1 AVSBench Datasets

In our work, a class-incremental audio-visual segmentation dataset (AVSBench-CI) is constructed from the well-known dataset AVSBench-semantic Zhou et al. (2024) to validate the proposed CMR. AVSBench-

Table 2: *mIoU* on the AVSBench-CI dataset for different class-incremental audio-visual segmentation scenarios.

| | 60-10 | | | | | | 60-5 | | | | | | 65-1 | | | | | |
|---|---|---|---|---|---|---|---|---|---|---|---|---|---|---|---|---|---|---|
| | Disjoint | | | Overlapped | | | Disjoint | | | Overlapped | | | Disjoint | | | Overlapped | | |
| Method | *1-60* | *61-71* | *all* | *1-60* | *61-71* | *all* | *1-60* | *61-71* | *all* | *1-60* | *61-71* | *all* | *1-65* | *66-71* | *all* | *1-65* | *66-71* | *all* |
| FT | 1.4 | 19.4 | 4.0 | 1.5 | 17.1 | 3.7 | 1.4 | 0.01 | 1.3 | 1.5 | 6.7 | 2.2 | 1.3 | 0.2 | 1.3 | 1.3 | 4.0 | 1.5 |
| LwF Rebuffi et al. (2017) | 10.1 | 25.1 | 12.3 | 7.1 | 19.0 | 8.8 | 1.5 | 9.7 | 2.6 | 1.5 | 12.6 | 3.0 | 1.3 | 0.7 | 1.3 | 1.3 | 4.5 | 1.6 |
| LwF-MC Li & Hoiem (2018) | 2.0 | 2.2 | 2.0 | 16.4 | 1.1 | 14.3 | 2.8 | 0.03 | 2.4 | 5.8 | 0.6 | 5.0 | 1.6 | 0.0 | 1.5 | 1.3 | 1.7 | 1.4 |
| ILT Michieli & Zanuttigh (2019) | 12.3 | 19.7 | 13.4 | 14.5 | 13.8 | 14.4 | 8.6 | 7.2 | 8.4 | 2.0 | 11.4 | 3.4 | 1.3 | 0.6 | 1.2 | 1.3 | 3.7 | 1.5 |
| MiB Cermelli et al. (2020) | 17.4 | 23.0 | 18.2 | 17.5 | 16.6 | 17.4 | 4.1 | 11.5 | 5.1 | 5.7 | 7.3 | 5.9 | 1.6 | 2.8 | 1.7 | 1.3 | 4.9 | 1.5 |
| PLOP Douillard et al. (2021) | 21.2 | 13.5 | 20.1 | 19.0 | 11.3 | 17.9 | 1.3 | 11.7 | 10.0 | 8.3 | 9.3 | 8.4 | 1.3 | 0.2 | 1.2 | 1.3 | 3.7 | 1.5 |
| AVSegFormer Gao et al. (2024) | 1.5 | 34.6 | 6.1 | 1.5 | 22.7 | 4.5 | 1.4 | 34.9 | 4.0 | 1.5 | 9.1 | 2.5 | 1.3 | 0.3 | 1.3 | 1.3 | 3.7 | 1.5 |
| EIR Yin et al. (2025) | 14.6 | 1.3 | 12.8 | 12.4 | 0.1 | 10.7 | 6.8 | 1.1 | 6.0 | 5.5 | 0.2 | 4.8 | 0.5 | 0.08 | 0.4 | 0.5 | 0.02 | 0.4 |
| CMR **(ours)** | **29.5** | **15.8** | **27.6** | **28.5** | **13.5** | **26.4** | **26.2** | **11.6** | **24.2** | **24.3** | **10.4** | **22.4** | **16.9** | **2.0** | **15.9** | **11.3** | **6.7** | **10.9** |
| Upper-bound | 33.7 | 33.2 | 33.7 | 34.3 | 29.6 | 33.7 | 33.7 | 33.2 | 33.7 | 34.3 | 29.6 | 33.7 | 34.0 | 28.7 | 33.7 | 34.0 | 29.8 | 33.7 |

semantic utilizes the techniques introduced in VGGSound Chen et al. (2020) to collect videos, ensuring that the audio and visual clips align with the intended semantics. The dataset provides semantic segmentation maps for videos as labels to enhance audio-visual semantic segmentation (AVSS). It contains a total of 11,356 videos spanning 70 categories. Each video segment consists of 10 frames of images and one 10-second audio clip. We divide the 70 categories in AVSBench-semantic for the original dataset into three training steps: 60-10, 60-5, and 65-1. Following the conventional continual semantic segmentation setup, the three training steps are divided into overlapped and disjoint settings to evaluate the model's performance under different task stream configurations.

In the overlapped setting, the classes are divided sequentially, meaning that classes from past and future tasks may appear in the current data and be labelled as background. In the disjoint setting, a community detection algorithm Sahu (2024) is employed to minimize the overlap of training data between consecutive steps. Therefore, the current data will not contain classes from future or past tasks. This setup closely aligns with continual learning scenarios. Furthermore, we expand the single-semantic dataset (AVSBench-CIS) and the multi-semantic dataset (AVSBench-CIM) based on the number of targets in the videos. AVSBench-CIS and AVSBench-CIM address scenarios involving modality entanglement with single-target and multi-target settings, respectively. The same settings are applied to these datasets.

## 4.2 Experimental Setup

### 4.2.1 Baselines

Since semantic segmentation can be regarded as a pixel-wise classification task, we compare our method with both classification and segmentation methods. Baselines, evaluation metrics, and implementation details are provided in the appendix.

For incremental classification baselines, we adapt LwF Li & Hoiem (2018), LwF-MC Rebuffi et al. (2017), and ILT Michieli & Zanuttigh (2019) to the audio-visual setting by replacing the visual-only backbone with our audio-visual encoder while keeping their continual learning strategies unchanged. For incremental segmentation baselines, we adapt MiB Cermelli et al. (2020), PLOP Douillard et al. (2021), and EIR Yin et al. (2025). All methods share the same backbone architecture (ResNet-50), and training schedule (30 epochs per task, batch size 2 per GPU on 4 NVIDIA 3090 GPUs, with the same optimizer and learning rate). The evaluation metric is *mIoU* (see Appendix for the formal definition).

## 4.3 Main Results

Tab. 2 illustrates the experiments of existing methods on AVSBench-CI. We use underlining to indicate the second-best performance. The upper bound represents the optimal performance when the model is directly trained on the target task. From left to right, task difficulty progressively increases, as more tasks lead to greater forgetting in the model. As reported in the results, our method achieves the best performance across all settings and demonstrates superior performance as the number of learning steps increases. On the more challenging 65-1 split, our method achieves signifcantly better performance than traditional approaches.

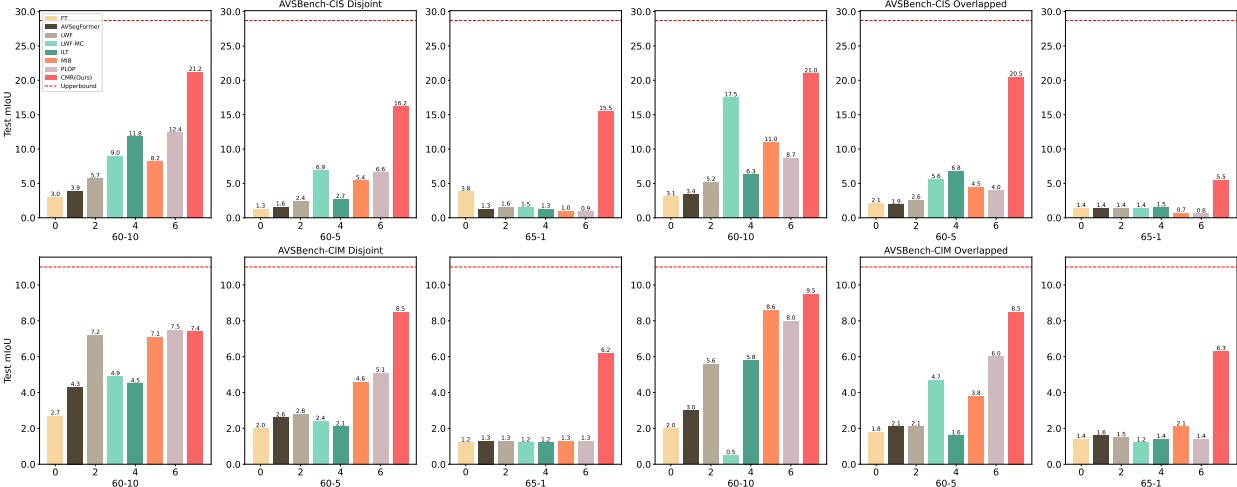

Figure 5: *mIoU* on the AVSBench-CIS and AVSBench-CIM datasets for different class-incremental audio-visual segmentation scenarios. The red line represents the upper bound. The upper section compares different methods and our method under different incremental settings on AVSBench-CIS, including both disjoint and overlapped scenarios. The lower section provides a similar comparison for AVSBench-CIM, showcasing the performance of our method.

Despite incorporating audio, traditional continual semantic segmentation suffers significant forgetting due to its inability to effectively disentangle audio-visual interactions. Specifically, EIR exhibits consistently low performance. The primary reason is the poor rehearsal quality resulting from its inability to extract audio aligned with the synthesized content, which exacerbates modality entanglement and consequently leads to catastrophic forgetting. Thus, experimental results show that disentangling modalities is essential in audio-visual segmentation to mitigate catastrophic forgetting.

Fig. 5 illustrates the experiments on AVSBench-CIS and AVSBench-CIM. Different colors represent different methods, and higher bars indicate better performance. The experimental results show that our method achieves a more significant improvement on AVSBench-CIS compared to AVSBench-CIM, with an increase of 11.3 *mIoU* on the AVSBench-CIS 60-10 overlapped setting, while only 1.5 *mIoU* on AVSBench-CIM. One main reason is that AVSBench-CIM can only select multi-target samples for rehearsal, which inherently involves dealing with the entanglement between multiple targets and modalities. In contrast, our observations indicate that single-target samples tend to yield better results when used for rehearsal. Therefore, for future work on multi-target tasks, it may be beneficial to preprocess the samples to enable the rehearsal of single-target samples. Nevertheless, our method achieves state-of-the-art performance on most tasks, demonstrating its effectiveness.

## 4.4 Ablation Study

### 4.4.1 Effectiveness of MSS and CSR

We evaluated the MSS against strategies based on maximum modality discrepancy, minimum modality discrepancy, and random sample selection. The results in rows 1-4 in Tab. 3 consistently demonstrate the superiority of the MSS. From Tab. 3, the further introduction of CSR based on MSS can further improve performance (e.g., 1.3% for the overlapped 1-60 setting), validating the effectiveness of CSR. Note that even random rehearsal improves over non-rehearsal baselines, which is expected and well-established in continual learning literature. The key contribution is that MSS improves over random selection by +2.0 *mIoU* (25.0 → 27.6 on disjoint "all"), and adding CSR further improves by +1.1 *mIoU*, demonstrating that both *what* you rehearse and *how frequently* you rehearse each class matter significantly. The uni-modal model required by MSS is a one-time overhead per task step (not per epoch), adding minimal computational cost.

Table 3: Ablation Study on effectiveness of MSS and CSR.

| | 60-10 | | | | | |
|---|---|---|---|---|---|---|
| | **Disjoint** | | | **Overlapped** | | |
| **Method** | *1-60* | *61-71* | *all* | *1-60* | *61-71* | *all* |
| Smallest | 25.6 | 13.1 | 23.7 | 21.8 | 12.7 | 20.5 |
| Largest | 25.2 | 14.6 | 23.8 | 23.4 | 12.3 | 21.9 |
| Random | 26.5 | 15.6 | 25.0 | 25.0 | 12.8 | 23.3 |
| MSS (Ours) | 28.7 | 13.4 | 26.5 | 27.2 | 13.2 | 25.3 |
| MSS+CSR **(Ours)** | **29.5** | **15.8** | **27.6** | **28.5** | **13.5** | **26.3** |

Table 4: Ablation Study on the number of rehearsal samples in MSS. We select 3, 5, and 7 samples per class using MSS.

| | 60-10 | | | | | |
|---|---|---|---|---|---|---|
| **Sample** | **Disjoint** | | | **Overlapped** | | |
| **Numbers** | *1-60* | *61-71* | *all* | *1-60* | *61-71* | *all* |
| MSS-3 | 27.3 | 14.7 | 25.6 | 25.5 | 12.2 | 23.6 |
| MSS-5 | 28.7 | 13.4 | 26.5 | 27.3 | 13.2 | 25.3 |
| MSS-7 | 28.0 | 13.3 | 25.9 | 29.3 | 12.7 | 26.9 |

## 4.5 Additional Ablation: Number of Rehearsal Samples

Tab. 4 reports the results of the ablation study on the number of rehearsal samples per class. The results show that as the number of rehearsal samples increases, the forgetting of old classes gradually decreases. However, an excessive number of rehearsal samples can inhibit learning new samples. Therefore, we select five samples per class for rehearsal.

## 4.6 Experiments on Transformer Architecture

To further validate the effectiveness of our method on Transformer-based architectures, we conduct additional experiments on the 60-10 and 60-5 settings using PVT (Pyramid Vision Transformer). The results in Tab. 5 demonstrate that our method continues to achieve competitive performance, even when applied to Transformer-based models, indicating its strong generalization capability across different architectural backbones.

Table 5: The results of our method on the AVSBench-CI 60-10 task based on PVT

| | 60-10 | | | | | |
|---|---|---|---|---|---|---|
| | **Disjoint** | | | **Overlapped** | | |
| **Backbone** | *1-60* | *61-71* | *all* | *1-60* | *61-71* | *all* |
| Ours (ResNet) | 29.5 | 15.8 | 27.6 | 28.5 | 13.5 | 26.3 |
| Ours (PVT) | 33.7 | 34.7 | 33.9 | 35.1 | 15.6 | 32.4 |

Table 6: The results of our method on the AVSBench-CI 60-5 task based on PVT.

| | 60-5 | | | | | |
|---|---|---|---|---|---|---|
| | **Disjoint** | | | **Overlapped** | | |
| **Backbone** | *1-60* | *61-71* | *all* | *1-60* | *61-71* | *all* |
| Ours (ResNet) | 26.2 | 11.6 | 24.2 | 24.3 | 10.4 | 22.4 |
| Ours (PVT) | 38.3 | 16.2 | 35.2 | 24.1 | 10.5 | 22.2 |

Moreover, results in Tab. 6 show that even with more powerful architectures, catastrophic forgetting still occurs under the 60-5 overlapped setting, highlighting the persistent challenge of knowledge retention in continual learning scenarios despite architectural advancements.

### 4.6.1 Qualitative Analysis of AVSBench-CI

Fig. 6 illustrates a qualitative comparison between our method and traditional methods. By replaying more samples from easily confused learned classes, our method enhances the ability of the model to leverage audio to distinguish between similar classes, thus effectively mitigating the misclassification between old and new classes. Furthermore, our model can segment learned classes such as airplanes, trains, and handpans, demonstrating superior semantic segmentation performance after learning new classes. Moreover, compared to existing methods, our method achieves more complete segmentation masks and yields finer details of the objects.

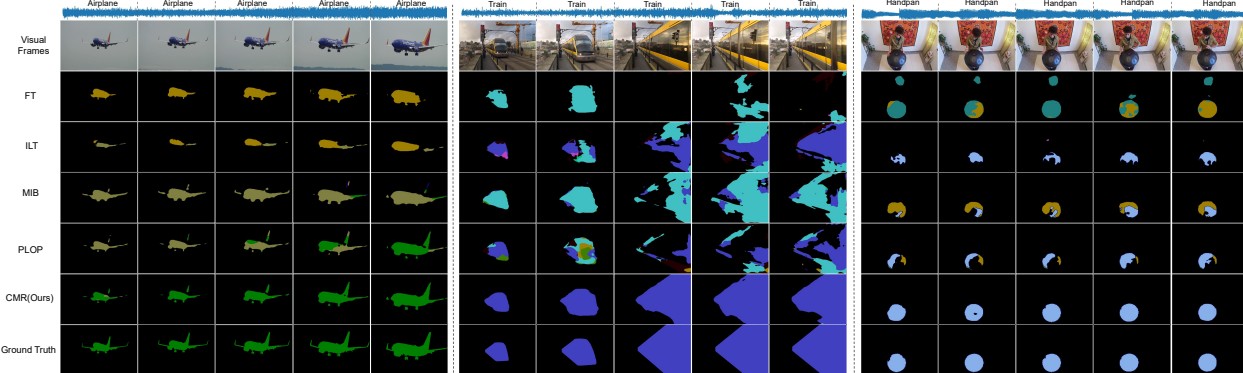

Figure 6: The qualitative results of incremental methods on the 60-10 setting of AVSBench-CI, where different colours represent different classes. The blue waveform represents the audio modality. Here, the far left represents the single old class (airplane), the middle represents the single new class (train), and the far right shows the sounding handpan (learned class) segmentation.

## 4.7 Comparison with Open-vocabulary AVS

While both open-vocabulary AVS Guo et al. (2024a) and continual learning AVS aim to enhance audio-visual segmentation, they address distinct challenges. Open-vocabulary AVS typically assumes access to all training data upfront and targets recognition of all categories during inference. In contrast, continual learning AVS deals with learning from a continuous data stream under memory constraints, without revisiting past data, which is a more realistic scenario for resource-limited or long-term deployments.

## 4.8 Analysis of Multi-modal Semantic Drift and Co-occurrence Confusion

To provide more direct evidence for the challenges discussed in Section 1, we present targeted analyses.

**Co-occurrence Confusion Analysis.** The collision pair statistics in Fig. 9 (Appendix) show that highly colliding categories correspond to objects that frequently co-occur (e.g., "guitar" and "man"). After applying CSR, the rehearsal frequency of these collision-heavy classes is increased, which directly reduces the confusion rate between co-occurring class pairs.

**Per-class Performance.** As shown in the qualitative results (Fig. 6), our method effectively segments previously learned classes such as "airplane" and "handpan" after learning new classes, while baseline methods exhibit significant confusion. The collision-based rehearsal strategy specifically improves performance on classes prone to co-occurrence confusion by increasing their rehearsal frequency proportionally to their collision frequency $\mathcal{F}_c$.

As shown in Tab. 7, open-vocabulary AVS achieves higher $mIoU$ on the base classes (15.9) but significantly lower performance on novel classes (8.6), whereas our CMR method achieves 35.3 $mIoU$ on novel classes through continual learning. This demonstrates that open-vocabulary AVS struggles to segment previously unseen classes in a sequential learning setting, while our continual learning approach can progressively learn to segment new classes effectively. The qualitative comparison in Fig. 7 further illustrates this difference.

Table 7: Quantitative comparison between Open-vocabulary AVS Guo et al. (2024a) and our CMR on the AVSBench-CI 60-10 disjoint setting. Base and Novel denote $mIoU$ on the base (1-60) and novel (61-71) classes, respectively.

| Method | Base | Novel |
|---|---|---|
| Open-vocabulary AVS Guo et al. (2024a) | 15.9 | 8.6 |
| CMR **(Ours)** | 5.2 | **35.3** |

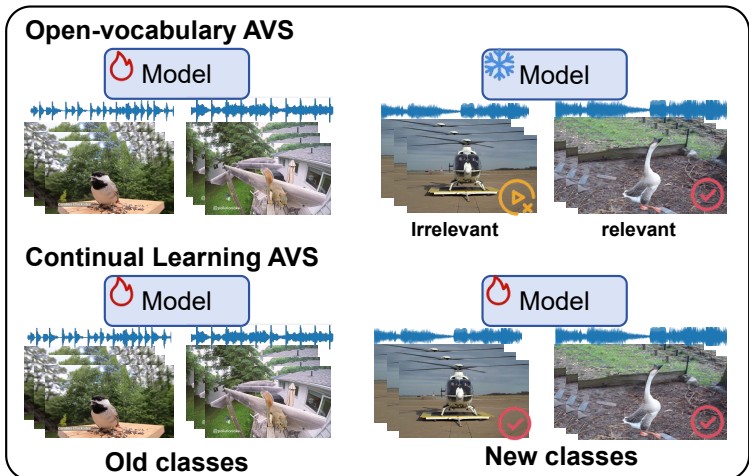

Figure 7: Qualitative comparison between Open-vocabulary AVS Guo et al. (2024a) and Continual Learning AVS. Open-vocabulary AVS fails to segment previously unseen objects, while our continual learning approach can progressively learn to segment new classes.

## 5 Conclusion

In this paper, we introduce a novel fine-grained multi-modal continual learning task: Continual Audio-Visual Segmentation. The task involves two critical challenges: multi-modal semantic drift and co-occurrence confusion. Through the collision-based multi-modal rehearsal framework, which includes a multi-modal sample selection and a collision-based sample rehearsal strategy, we mitigate the incorrect modality semantic associations caused by these two challenges. Comprehensive experiments demonstrate the effectiveness of our method. The current CAVS setup assumes fixed-length 10-second video clips inherited from the AVSBench benchmark. Extending CAVS to long-video settings, where audio-visual alignment varies temporally and multi-modal semantic drift may be more severe, is an important direction for future work.

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

# A  Appendix

## A.1  Multimodal Learning

Multimodal learning Wei et al. (2024b); Wei & Hu (2024); Xiu et al. (2025); Guo et al. (2024b) focuses on integrating information across diverse modalities and investigating the intricate interrelationships between

them in various contexts. Wei Wei et al. (2024b) first estimates each modality's learning status based on separability in its unimodal representation space, then uses this to softly initialize the corresponding unimodal encoder. MMPareto Wei & Hu (2024) employs gradient-based optimization to mitigate model bias towards specific modalities during training, thereby enhancing multimodal learning performance. To compare with more recent work, Finger Xiu et al. (2025) focuses on distinguishing foreground from background and transferring unimodal knowledge, while we focus on selecting consistent samples through modal contribution and replaying them according to collision frequency. From task level, Finger aims to seamlessly integrate new classes with limited incremental samples, while we focus on avoiding interference with old task knowledge when training on new tasks. Meanwhile, in contrast to Open-set AVS, continual learning AVS deals with learning from a continuous data stream under memory constraints, without revisiting past data.

## A.2 Evaluation Metrics

Following Cermelli et al. (2020), mean Intersection-over-Union ($mIoU$) is taken for evaluation:

$$mIoU = \frac{1}{N} \sum_{i=1}^{N} \frac{TP_i}{TP_i + FP_i + FN_i}, \tag{11}$$

where $TP_i$ denotes the number of samples correctly predicted as $class_i$, $FP_i$ represents incorrectly predicted as $class_i$, $FN_i$ indicates the number of samples that the model failed to correctly predict as $class_i$.

## A.3 Implementation Details

Our method builds upon the best-performing PLOP model combined with the memory. We have primarily conducted training and evaluation using ResNet-50 He et al. (2016) pre-trained on ImageNet Deng et al. (2009). The ASPP module Zhou et al. (2022) is utilized as the fusion module. For input frames, we resize the resolution to $224 \times 224$. The same data augmentation is applied as in Zhou et al. (2022), excluding memory data. The training batch size is set to 2 per GPU on 4 Nvidia L40 48GB GPUs, and the training runs 30 epochs per task. For single-modal training, all steps are trained only using visual-modal data. For memory samples, 5 samples per class are selected for rehearsal, and the number of resampled samples is set to 20% of the total sample size. To be fair, all tasks share a common test set with all learned classes.

## A.4 Baseline Methods

For incremental classification methods: (1) Learning without forgetting (LWF) Li & Hoiem (2018): LWF distils the output differences between the old and current models. Our implementation of LwF follows Li & Hoiem (2018); distillation and cross-entropy losses share the same label space and classifier. (2) LwF multi-class (LWF-MC) Rebuffi et al. (2017): LwF-MC utilizes multiple binary classifiers. Following the approach proposed in Cermelli et al. (2020), LWF-MC is implemented by combining two binary cross-entropy losses in a weighted manner. These losses are computed based on the ground truth labels and the probabilities predicted by the previous model $f_{\theta_{t-1}}$. (3) ILT Michieli & Zanuttigh (2019): ILT employs a dual-space knowledge distillation strategy, including a distillation loss in the output space and an additional distillation loss in the feature space.

For incremental segmentation: (1) MiB Cermelli et al. (2020): MiB uses complete output space distillation and background uncertainty propagation. (2) PLOP Douillard et al. (2021): PLOP proposes multi-scale pooling distillation to maintain spatial relationships at the feature level and uses entropy-based pseudo-labels to annotate background classes predicted by the old model. (3) EIR Yin et al. (2025) is an instance rehearsal method for continual semantic segmentation, introduced in CVPR 2025, and represents the state-of-the-art (SOTA) in this field. In our work, we reproduced both the original EIR method and its PLOP-based variant, and adapted them to the continual audio-visual segmentation. Our experiments demonstrate that the PLOP-enhanced EIR outperforms the vanilla EIR approach. To ensure a fair comparison, we adopt the PLOP-based EIR method in our study. Besides, the fine-tuning of AVSegFormer Gao et al. (2024) is implemented based on ResNet-50. Additionally, fine-tuning each task as a baseline and offline training on all classes is provided as an upper bound for performance comparison.

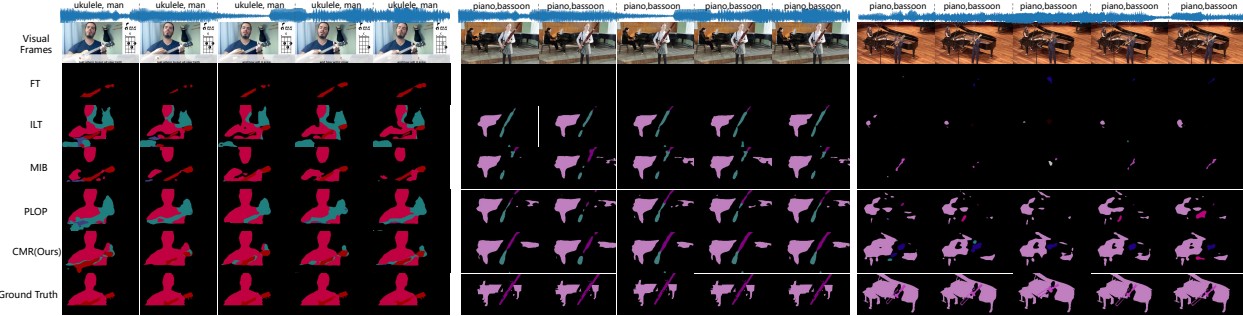

Figure 8: We demonstrate the comparative performance of our method on the AVSBench-CIM dataset, where multiple objects often emit sounds simultaneously, thereby placing higher demands on the model's ability to perform continuous audio-visual segmentation. Visualization studies on the AVSBench-CIM dataset demonstrate that our method consistently achieves robust and superior performance in complex scenarios containing multiple co-occurring objects.

Table 8: The table presents the 60-10 category configuration of AVSBench under the disjoint setting.

| Disjoint Settings | AVSBench-CI |
|---|---|
| 60-10 step 0 | erhu, cello, bus, airplane, parrot, bassoon, missile-rocket, accordion, goose, hen, baby, horse, saxophone, boat, frying-food, flute, marimba, bird, hair-dryer, harmonica, mower, emergency-car, tiger, saw, duck, squirrel, clarinet, dog, guitar, keyboard, boy, clipper, handpan, sitar, elephant, tabla, girl, gun, axe, harp, piano, car, guzheng, drum, helicopter, motorcycle, clock, man, tank, train, sorna, sheep, lion, leopard, pipa, bell, tractor, pig, donkey, cat |
| 60-10 step 1 | wolf, tuba, trumpet, utv, violin, ukulele, trombone, vacuum-cleaner, woman, truck |

Table 9: Number of collision pairs. Highly colliding categories typically correspond to objects that co-occur. The categories with the highest collision rate are "guitar" and "man," which aligns with real-world observations.

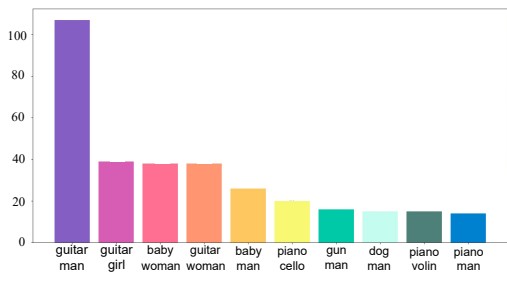

## A.5 Additional Qualitative Results

### A.5.1 Qualitative Analysis of AVSBench-CIM

Fig. 8 demonstrates a comparison between our method and previous methods on AVSBench-CIM, highlighting the superior performance of our method in scenarios requiring the segmentation of multiple targets. The figure presents three multi-target cases. In the first case, where the goal is to segment "ukulele" and "man," our method achieves complete segmentation of both objects compared to previous methods while exhibiting significantly less class confusion. In the third case, while previous methods fail to segment the target object entirely, our method successfully segments most of the "piano." These examples further prove the superiority of our method in multi-target audio-visual segmentation tasks.

### A.5.2 The Details of Category

Tab. 8 present the 60-10 category learning sequence under the setting of disjoint in the AVSBench-CI dataset. For the setting of disjoint, we employ the Louvain algorithm to divide the 70-category dataset into bipartite and tripartite graphs. Classes with minimal overlapped are then allocated to distinct steps to form the

disjoint dataset. The dataset was directly partitioned into steps based on sequential category order for the overlapped setting.

### A.5.3 Qualitative Analysis of Collision Classes

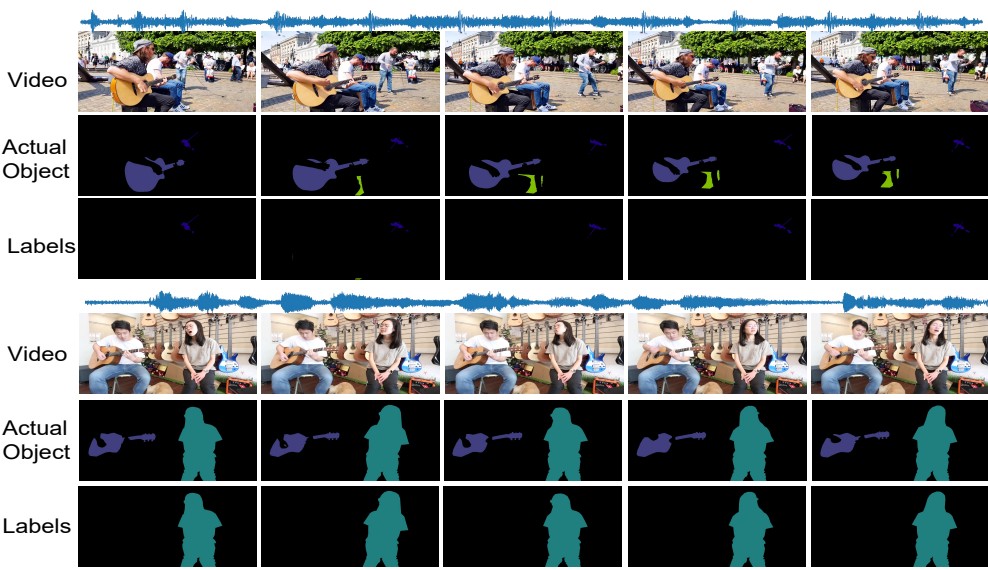

Figure 9: Example of Multi-modal semantic drift. The image illustrates the phenomenon of multi-modal semantic drift.

Experimental observations indicate that collision classes frequently co-occur in previous tasks, leading the model to perceive these classes as semantically similar. The statistics on the number of collision pairs in Fig. 9 validate our hypothesis. This phenomenon occurs because the model lacks prior semantic knowledge of new classes and tends to associate frequently co-occurring targets with similar features. Consequently, the forgetting process in continual learning can be viewed as the model correcting this cognitive bias after learning new classes, which often leads to catastrophic forgetting.

### A.5.4 Examples of Multi-modal Semantic Drift

To better understand the Multi-modal semantic drift task, we present two examples from the AVSBench-CI 60-10 task. The classes "guitar" and "drum" were learned in step 0, while "violin" and "woman" are to be learned in step 1. During the learning process of step 1, "guitar" and "drum" are labeled as background. This causes their corresponding audio to be associated with background semantics, leading to the multi-modal semantic drift.

### A.5.5 Effect Analysis of Multi-modal Sample Selection (MSS)

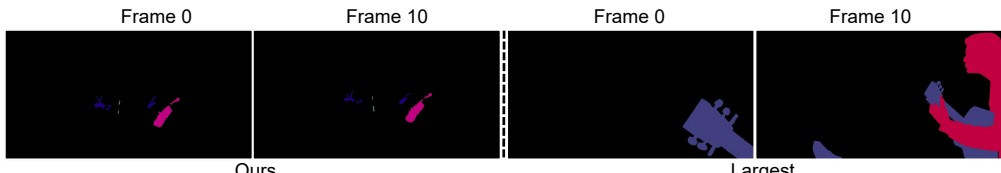

Figure 10: Comparison of sample selection strategy. The image visualizes our method alongside the sample selection strategy based on maximum modality discrepancy, where samples selected by MSS exhibit greater consistency.

As shown in Fig. 10, the samples selected by MSS exhibit the following characteristics: (1) Unlike samples with multiple targets, MSS tends to favour samples with single targets. (2) MSS prefers samples where the target is consistently present. (3) MSS prioritizes samples with better alignment between the target audio and visual modalities. This phenomenon aligns with our initial hypothesis, as these three types of samples typically exhibit less multi-modal semantic drift, thereby aiding the model in better retaining knowledge of old classes.

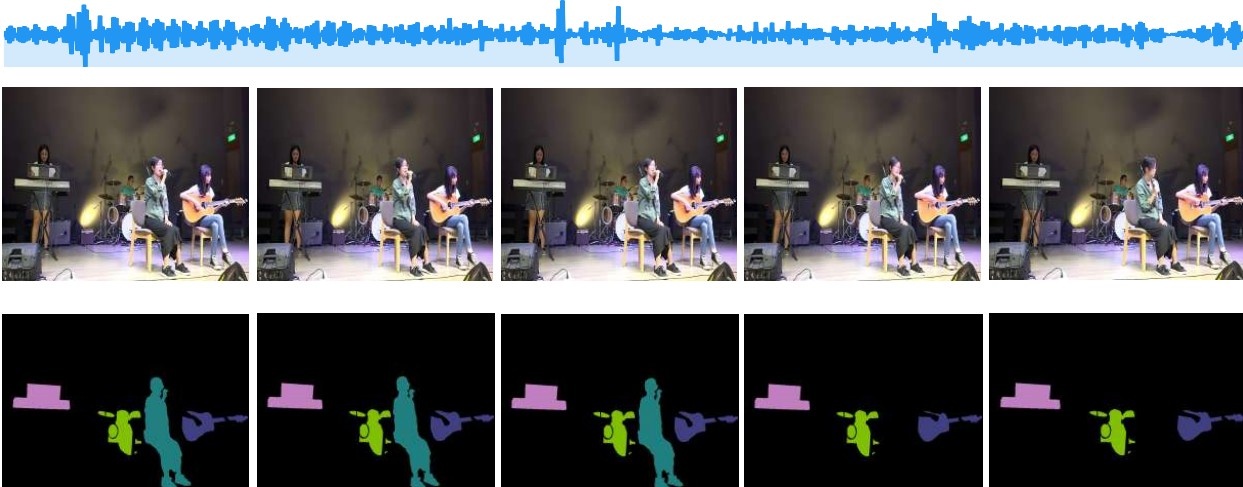

Figure 11: Failure cases of CMR. Examples where the CMR fails, including visually similar classes and multi-target ambiguity.

### A.6  Failure Case Analysis

To provide a more balanced evaluation, we present failure cases of our CMR framework in Fig. 11. Despite the effectiveness of MSS and CSR, our method still struggles in certain challenging scenarios, such as visually similar classes with distinct audio signatures where pixel-level confusion persists, or multi-target scenes where multiple sounding objects produce ambiguous audio signals. These cases highlight the limitations of the current framework and motivate future research directions.

### A.7  Discussion on Long-Video Applicability

The current CAVS setup inherits the 10-second, 10-frame assumption from the AVSBench benchmark. In real-world embodied intelligence scenarios, videos are typically much longer and audio-visual alignment varies temporally. We discuss the applicability and limitations of CMR when extended to long-video settings.

The core principles of CMR (modality-consistent sample selection and collision-based rehearsal) are not inherently limited to fixed-length inputs. MSS computes $\Delta(S_a)$ as an average over frames, which naturally extends to variable-length sequences. CSR operates on collision statistics aggregated across samples, independent of individual video length.

However, several challenges arise in long-video settings: (1) Multi-modal semantic drift may be more severe, as the audio signal may correspond to different sounding objects at different timestamps within a single video. (2) The $mIoU$-based modality contribution metric may need to be computed at the segment level (temporal windows) rather than the full video level to capture local audio-visual alignment. (3) Memory buffer design may need adaptation, as storing full long videos is memory-intensive, so segment-level selection may be more practical.

Extending CAVS to long-video settings is an important future direction. CMR's components could be adapted through segment-level MSS and temporally-aware collision detection to handle the more complex temporal dynamics in long videos.

