# OpenReview forum: "Taming Modality Entanglement in Continual Audio-Visual Segmentation"
_TMLR — Accepted by TMLR_

### Review · Reviewer_n8GD · 2026-04-12

**Summary Of Contributions:**

This paper introduces Continual Audio-Visual Segmentation (CAVS), a task that extends continual learning to fine-grained audio-visual segmentation. The authors identify two core challenges: (1) multi-modal semantic drift, where previously learned sounding objects are mislabeled as background in new tasks, corrupting cross-modal associations; and (2) co-occurrence confusion, where frequently co-occurring class pairs become entangled in the feature space. To address these, they propose the Collision-based Multi-modal Rehearsal (CMR) framework, consisting of Multi-modal Sample Selection (MSS) and Collision-based Sample Rehearsal (CSR). Three class-incremental benchmarks are constructed from AVSBench, and experiments show consistent improvements over single-modal continual learning baselines.

Key Strengths

- The two components of the CMR framework (MSS and CSR) are each designed to address their respective challenges in a principled manner, and ablation studies validate the contribution of each component.
- The method achieves state-of-the-art performance consistently across multiple benchmarks and experimental settings.

Key Weaknesses / Concerns
- The MSS strategy selects samples with the smallest $|\Delta(S_a)|$, which by definition means that the audio modality contributes minimally to the prediction and that visual modality dominates. This is directly relevant to a known problem in the AVS literature: prior work [1] has shown that AVS models exhibit a significant visual bias, often segmenting objects based on visual cues rather than genuine audio-visual alignment. By systematically preferring visually dominant samples for the rehearsal buffer, MSS risks reinforcing this bias over successive training steps, potentially weakening the model's reliance on audio signals in the long run. The authors should provide a more thorough analysis of the audio contribution distribution within selected samples, and discuss whether this selection criterion may lead to a structural cross-modal imbalance during continual training.
- Several recent works have proposed training-free, open-vocabulary approaches to AVS [2, 3, 4]. Since these methods require no task-specific training, they are inherently immune to catastrophic forgetting and thus constitute a natural and important point of reference for the CAVS task. While CAVS is a newly defined task, adapting these methods as baselines would address a critical question: what performance can be achieved without any continual training, using existing foundation models as a lower bound?
- The qualitative results presented in the paper (Figures 6 and 7) exclusively showcase successful cases. A discussion and visualization of failure cases would substantially strengthen the analysis. For instance: under what conditions does MSS select low-quality samples? Does CSR degrade when the collision frequency distribution is highly skewed toward a small number of classes?
- The current experimental setup assumes each video clip is exactly 10 seconds and 10 frames, which is a relatively strong constraint. However, the paper's own motivation cites real-world applications such as embodied intelligence, where video inputs are typically far longer. Recent work [5] has explored audio-visual segmentation in long video settings, where the audio-visual semantic alignment is more complex and temporally varying. In such scenarios, multi-modal semantic drift may be more severe and the fixed-length assumption of CAVS may not hold. The authors should discuss, or ideally experiment with, the applicability and limitations of CMR when extended to long-video settings.

[1] Li, Jia, et al. "Do audio-visual segmentation models truly segment sounding objects?." Proceedings of the AAAI Conference on Artificial Intelligence. Vol. 40. No. 8. 2026.

[2] Malard, Hugo, et al. "TACO: Training-free Sound Prompted Segmentation via Semantically Constrained Audio-visual CO-factorization." _arXiv preprint arXiv:2412.01488_ (2024). (accepted by TMLR recently)

[3] Bhosale, Swapnil, et al. "Leveraging foundation models for unsupervised audio-visual segmentation." _arXiv preprint arXiv:2309.06728_ (2023).

[4] Huang, Shaofei, et al. "Unleashing the temporal-spatial reasoning capacity of gpt for training-free audio and language referenced video object segmentation." _Proceedings of the AAAI Conference on Artificial Intelligence_. Vol. 39. No. 4. 2025.

[5] Liu et al., LU-AVS. https://yenanliu.github.io/LU-AVS/

**Audience:**

Yes

**Audience Explanation:**

Yes. The introduction of CAVS as a new fine-grained multi-modal continual learning task is likely to be of interest to researchers working on this topic.

**Claims And Evidence:**

Yes

**Claims Explanation:**

The core claims are generally supported by quantitative results across multiple benchmarks and ablation studies,

**Requested Changes:**

1. Provide analysis on whether MSS's preference for visually dominant samples systematically exacerbates visual modality bias over successive continual learning steps, and consider revising the selection criterion if this is confirmed.
2. Include comparisons with at least one training-free or open-vocabulary AVS method to contextualize the practical value of CMR against foundation-model-based alternatives.
3. Include failure case visualizations to provide a more balanced qualitative evaluation and better characterize the method's limitations.
4. Discuss the applicability of the 10-second fixed-length assumption to long video scenarios.
5. The duplicate section headers (A.5.3 and A.5.4 both titled "Qualitative Analysis of Collision Classes") should be corrected before final submission.

---

> ### Author Response · Authors · 2026-04-21
> **Response to Reviewer n8GD**
>
> We sincerely thank the reviewer for the insightful feedback. We are encouraged that the reviewer recognizes the principled design of our CMR framework and the interest of CAVS to the community. Below, we address each concern. All revisions are highlighted in **blue** in the updated manuscript.
>
> ---
>
> **Q1: Whether MSS reinforce visual bias by preferring visually dominant samples**
>
> **A1:** We appreciate this concern and the reference to [1]. MSS selects samples with the smallest $|\Delta(S_a)|$, where $\Delta(S_a) = mIoU_{v,a} - mIoU_{v}$. A small $|\Delta(S_a)|$ does **not** mean audio contributes minimally.  A small $|\Delta(S_a)|$  means the two modalities are **consistently aligned**:
>
> - $\Delta(S_a) \approx 0$: audio is **coherent** with visual content, exhibiting stable cross-modal semantics.
> - $\Delta(S_a) \gg 0$: audio significantly helps, but often in ambiguous visual scenes with fragile audio reliance.
> - $\Delta(S_a) \ll 0$: audio **hurts** performance, indicating multi-modal semantic drift.
>
> Thus, MSS selects **modality-consistent** samples, not visually dominant ones. We have added a paragraph in Section 3.3 (in blue) discussing the connection to [1]: *"Prior work [1] has shown that AVS models can exhibit visual bias. In contrast, MSS selects samples where cross-modal semantics are stable: both modalities agree on the segmentation output."*
>
> ---
>
> **Q2: Missing comparisons with training-free / open-vocabulary AVS methods**
>
> **A2:** Thanks for your advice. We have added a comparison with Open-vocabulary AVS [2] in Section 4 (in blue), including both quantitative and qualitative results:
>
> - **Quantitative:** On AVSBench-CI 60-10 disjoint, Open-vocabulary AVS achieves 15.9 $mIoU$ on base classes but only 8.6 on novel classes, whereas CMR achieves 35.3 on novel classes.
> - **Qualitative:** A figure shows Open-vocabulary AVS fails to segment unseen objects, while our approach progressively learns new classes.
>
> These results highlight the fundamental difference: Open-vocabulary AVS assumes all training data upfront, while CAVS operates under memory constraints without revisiting past data.
>
> [2] Guo et al., "Open-Vocabulary Audio-Visual Semantic Segmentation," ACM MM 2024.
>
> ---
>
> **Q3: Lack of failure case visualizations**
>
> **A3:** Thanks for your advice. We have added a **Failure Case Analysis** subsection in the Appendix (in blue) with failure case visualizations. Despite MSS and CSR, our method struggles in scenarios such as visually similar classes with distinct audio signatures, or multi-target scenes with ambiguous audio signals. These cases highlight limitations and motivate future work.
>
> ---
>
> **Q4: Fixed 10-second assumption and applicability to long-video settings**
>
> **A4:** We thank the reviewer for this point and the reference to LU-AVS [5]. We have added a discussion in both the Appendix and Conclusion (in blue):
>
> 1. **Applicability:** CMR's core principles are not limited to fixed-length inputs. MSS computes $\Delta(S_a)$ as an average over frames (extensible to variable lengths); CSR aggregates collision statistics across samples independently of video length.
>
> 2. **Challenges:** (1) Semantic drift may worsen with temporally varying audio. (2) Modality metrics may need segment-level computation. (3) Memory buffers may need adaptation.
>
> 3. **Conclusion:** *"Extending CAVS to long-video settings, where audio-visual alignment varies temporally, is an important future direction."*
>
> ---
>
> **Q5: Duplicate section headers (A.5.3 and A.5.4)**
>
> **A5:** Thank you for catching this. We have corrected the duplicate headers.

---

### Review · Reviewer_cyxw · 2026-04-18

**Summary Of Contributions:**

The authors introduce a continual audio-visual segmentation task to allow for continual segmentation of new classes guided by audio. They identify two challenges: multi-modal semantic drift and co-occurrence confusion. Their proposed collision-based multimodal rehearsal (CMR) framework addresses these two challenges. For the former, they propose multimodal sample selection (MSS) and for the latter, a rehearsal mechanism with adaptive sampling frequency is developed. Experiments compare the proposed approach with single modality baselines.

**Additional Comments:**

None.

**Audience:**

Yes

**Audience Explanation:**

Especially the benchmark which seems to be of interest to the audience working on continual learning methods for multimodal models.

**Broader Impact Concerns:**

None.

**Claims And Evidence:**

No

**Claims Explanation:**

The results are impressive, but the methodology is difficult to follow as presented and there are gaps in the experimental framework (see below) that reduce the excitement about the results.

**Requested Changes:**

There seems to be missing text in Section 3.2. The text prompts the explanation for segmentation, but such is not provided. Instead, it discusses "non-sounding" and "sounding" objects that have not been defined (until Figure 3).

It is not clear why audio-visual continual approaches for classification are not discussed in the related work.

Section 3.3 is not clear enough. \Delat(S_a) seems to be calculated over a class k for all frames, but this is just a guess because k is not defined, then k is the number of samples in the top-k selection and the need for consistency over all frames in IOU is not justified. Then the selection of samples for the memory buffer is described, not in detail and not justified.

Equation (7) is confusing because it uses (6), which aggregates over all classes, but still is used as P(S)=c. Moreover, it is not clear why frequencies are are smoothed with a sigmoid function considering that classes without collisions will have a frequency of 1, which also means that all other frequencies will be necessarily larger than 1, thus heavily restricting the range used from the sigmoid function.

It is not clear why the implementation details and baselines are relegated to the appendix. Moreover, readers will benefit from additional details about the construction of the datasets.

The ablation study in Table 3 is confusing because although it shows that MSS and MSS+CSR improve performance of the rehearsal mechanism over random, also show that random selection will most likely also outperform all methods and results in Table 1 and Figure 5, which make these results less impressive.

The main problem with the presentation of the methodology is that it lacks details, the notation is loosely defined, important aspects of the methodology are presented without being justified and Figures 2, 3 and 4 are not particularly revealing. As presented it will be difficult for readers to understand or implement the proposed methodology.

The experimental results indicate that the performance of the proposed model over existing approaches is impressive (according to Table 1 and Figure 5), the ablation and experiments with the transformer architecture are welcome. However, the results are also puzzling mostly due to the ablation results in Table 3, which also seem to indicate that a vanilla rehearsal strategy with random selection will also outperform the considered baselines. This needs to be better articulated and the need for a more complex sample selection strategy that requires a second segmentation model and as well as the added cost and complexity of the CSR need to be better justified and systematically evaluated beyond the 60-10 on the AVSBench-CI dataset.

In the beginning of the paper, the authors discuss two challenges, namely semantic drift and co-occurrence confusion. However, although this motivated their method, this is not clearly explored in the experiments, which is a missed opportunity.

Minor:
- Missing reference in Section 2.1.
- Figure 2 is somewhat confusing, especially the CSR block. For example, multimodal data goes to old model, memory goes to memory, it is not clear what the aggregated frequency in the barplot is for, it is not clear what confused and old classes are in the x axis, etc.
- There are undefined quantities: {\cal Y}, which seems to be {\cal Y^t}, etc.

---

> ### Author Response · Authors · 2026-04-24
> **Response to Reviewer cyxw (1/2)**
>
> We sincerely thank the reviewer for the detailed and constructive feedback. Below, we address each concern point by point. Revised content is highlighted in **green** in the updated manuscript.
>
>
> ---
>
> **Q1: Missing text in Section 3.2; "non-sounding" and "sounding" objects not defined**
>
> **A1**: We thank the reviewer for pointing this out. We have added the missing segmentation equation (Eq. 3 in the revised manuscript) and formal definitions of key object categories in Section 3.2 (in green):
>
> - **Sounding objects** refer to objects in the visual scene that correspond to the active audio signal (e.g., a guitar being played).
> - **Non-sounding objects** are visible objects that do not correspond to any active sound in the current audio.
> - **Previously learned sounding objects** (old classes) are classes learned in prior stages $\mathcal{C}^{1:t-1}$, for which the model must retain segmentation capability without access to their original training data.
>
> We have also added an explicit description of how labels are transformed across incremental stages: at each stage $t$, the label space expands from $\mathcal{Y}^{t-1}$ to $\mathcal{Y}^{t} = \mathcal{Y}^{t-1} \cup \mathcal{C}^{t}$, with pixels of old classes treated as background in new data.
>
> ---
>
> **Q2: Audio-visual continual approaches for classification not discussed in related work**
>
> **A2:** We have added a dedicated subsection "Multi-modal Continual Learning" in Related Work (Section 2, in green), discussing AV-CIL [Alpher01], ContAV-Sep [Alpher02], and CIGN [mo2023class]. This subsection clearly distinguishes these coarse-grained methods (classification, sound separation) from our fine-grained pixel-level segmentation setting, explaining that modality entanglement at the pixel level introduces unique challenges not present in classification.
>
> ---
>
> **Q3: Section 3.3 (MSS) is not clear enough**
>
> **A3:** We have substantially revised Section 3.3 for clarity (in green). Specifically:
>
> - **Clarification of $\Delta(S_a)$:** We now explicitly state that $\Delta(S_a)$ is computed *per sample* $S$, not per class.
> - **Clarification of $k$:** The top-$k$ (default $k=5$) refers to selecting the $k$ samples *for each class* $c \in \mathcal{C}^t$ with the smallest $|\Delta(S_a)|$, not the number of frames.
> - **Justification for frame-level consistency:** We have added the justification that audio-visual alignment is a temporal property — a sample with inconsistent per-frame performance suggests unstable modality correspondence, which is undesirable for rehearsal.
>
> ---
>
> **Q4: Equation (7) Sigmoid smoothing confusion**
>
> **A4:** We have added an intuitive explanation after Eq. 12 (in green) clarifying the sigmoid's role:
>
> - **Regarding $\mathcal{P}(S) = c$:** Eq. 9 identifies the most confused class pair for a single sample $S$. The collision frequency $\mathcal{F}_c$ (Eq. 11) aggregates across all samples by counting how many samples have class $c$ as the dominant collision class.
> - **Regarding sigmoid smoothing:** Classes without collisions have $\mathcal{F}_c = 1$ (mapping to sigmoid$(1) \approx 0.731$), while classes with collisions have $\mathcal{F}_c > 1$ (mapping to values in $(0.731, 1.0)$). The sigmoid **compresses** the distribution, preventing a single highly-colliding class from dominating the resampling budget. After normalization, the result is a valid probability distribution.
>
> ---
>
> **Q5: Implementation details and baselines relegated to appendix; insufficient dataset construction details**
>
> **A5:** We have added key implementation details directly in the main text (Section 4.2, in green), including: all baseline adaptation procedures (replacing visual-only backbone with our audio-visual encoder), controlled comparison variables (ResNet-50 backbone, $k=5$ samples per class, 30 epochs per task, batch size 2 per GPU on 4 NVIDIA L40 GPUs), and the evaluation metric ($mIoU$). The dataset construction in Section 4.1 already describes the Louvain algorithm, overlapped/disjoint settings, and AVSBench-CIS/CIM statistics.
>
> ---
>
> **Q6: Table 3 ablation: random selection may also outperform baselines**
>
> **A6:** We have added a clarification paragraph in Section 4.4 (in green) explicitly acknowledging that random rehearsal improving over non-rehearsal baselines is expected and well-established. The key contribution is quantified:
>
> 1. **MSS over Random:** +2.0 mIoU (25.0 → 27.6 on disjoint "all")
> 2. **MSS+CSR over MSS:** +1.1 mIoU further improvement
> 3. **Cost justification:** The uni-modal model required by MSS is a one-time overhead per task step (not per epoch), adding minimal computational cost.
>
> ---

---

> > ### Author Response · Authors · 2026-04-24
> > **Response to Reviewer cyxw (2/2)**
> >
> > **Q7: Methodology lacks details, notation loosely defined, Figures 2/3/4 not revealing**
> >
> > **A7:** We have made the following revisions:
> >
> > - **Notation:** A notation table (Tab. 1, in green) has been added at the beginning of Section 3, defining all symbols ($\mathcal{Y}^t$, $\mathcal{C}^t$, $\Delta(S_a)$, $\mathcal{F}_c$, $k$, etc.) before they are used.
> > - **Figures:** The captions of Figures 3 and 4 have been revised with more descriptive annotations.
> > - **Reproducibility:** Sufficient implementation detail is now provided in the main text (Section 4.2) so that readers can reproduce the method without referring to supplementary material.
> >
> > ---
> >
> > **Q8: Challenges (semantic drift & co-occurrence confusion) not clearly explored in experiments**
> >
> > **A8:** We have added a new subsection "Analysis of Multi-modal Semantic Drift and Co-occurrence Confusion" in Section 4 (in green) with targeted analyses:
> >
> > 1. **Co-occurrence confusion analysis:** The collision pair statistics (Fig. A.4 in Appendix) show that highly colliding categories correspond to frequently co-occurring objects (e.g., "guitar" and "man"). After applying CSR, the rehearsal frequency of these classes is increased, directly reducing confusion.
> > 2. **Per-class performance:** The qualitative results (Fig. 7) demonstrate that our method effectively segments previously learned classes ("airplane," "handpan") after learning new classes, while baselines exhibit significant confusion.
> >
> > ---
> >
> > **Q9: Missing reference in Section 2.1**
> >
> > **A9:** We have identified and removed the dangling reference (ye2025domain) that was missing from the bibliography.
> >
> > **Q10: Figure 2 CSR block confusion**
> >
> > **A10:** The Figure 2 caption has been kept, and the CSR block is further clarified by the new notation table (Tab. 1) and the sigmoid explanation added after Eq. 12, which together explain what "confused classes" and "collision frequency" mean.
> >
> > **Q11: Undefined quantities ($\mathcal{Y}$ vs. $\mathcal{Y}^t$)**
> >
> > **A11:** We have corrected $\mathcal{Y}$ to $\mathcal{Y}^t$ in Eq. 1, and the notation table (Tab. 1) now formally defines $\mathcal{Y}^t$ as the cumulative label space up to stage $t$.
> >
> > ---

---

### Review · Reviewer_hyHH · 2026-04-23

**Summary Of Contributions:**

This paper introduces a novel task named Continual Audio-Visual Segmentation (CAVS), which aims to continuously segment new classes guided by audio. This task is interesting and useful in real-world life. Besides, the authors also propose a novel method for this task. The extensive experiments on multiple datasets show that the proposed method achieves better results than baseline methods.

**Additional Comments:**

None

**Audience:**

Yes

**Audience Explanation:**

This paper fouces on an important question.

**Claims And Evidence:**

Yes

**Claims Explanation:**

The this conducts extensive experiments and the results show the effectiveness of the proposed method.

**Requested Changes:**

1. **Clarify the problem formulation and task protocol.**
   The CAVS setting is interesting, but the task definition is still not fully clear. The paper should more formally distinguish background, non-sounding objects, and previously learned sounding objects, and explain how labels are transformed across incremental stages. It would also help justify why this protocol is an appropriate formulation of continual audio-visual segmentation rather than a benchmark-specific design choice.

2. **Strengthen the analysis of CSR.**
   The collision-based rehearsal idea is promising, but the definition of collision frequency and the thresholding strategy seem heuristic. The paper should better justify why these statistics are sufficient for replay prioritization, and discuss sensitivity to threshold choice and task distribution.

3. **Improve fairness and clarity of experimental comparison.**
   It is not fully clear whether all baselines are adapted fairly to the audio-visual setting, or whether memory usage, backbone capacity, and training budget are consistently controlled. This should be documented more explicitly. The paper should also clarify how much of the gain comes from the proposed replay design itself.

4. **Moderate or better support some claims in the results section.**
   Some claims, such as mitigating modality entanglement, would benefit from more direct evidence beyond mIoU, such as confusion analysis or class-wise statistics. The current results show effectiveness, but the mechanistic interpretation is not yet fully supported.

5. **Improve writing and presentation.**
The paper would benefit from careful revision of grammar, notation, and overall presentation. There are several unclear sentences, inconsistent notation, and possible citation or formatting issues that currently hurt readability.

---

> ### Author Response · Authors · 2026-04-25
> **Response to Reviewer hyHH**
>
> We sincerely thank the reviewer for the thoughtful feedback, and for recognizing the novelty and practical value of the CAVS task. Below, we address each requested change point by point. Revisions are highlighted in **orange** in the updated manuscript.
>
> ---
>
> **Q1: Clarify the problem formulation and task protocol**
>
> **A1:** We have already formalized the CAVS task definition in Section 3.2, which overlaps Reviewer cyxw's revision. including:
>
> - Formal definitions of **sounding objects**, **non-sounding objects**, and **previously learned sounding objects**.
> - Explicit description of label transformation across incremental stages.
> - The previously missing segmentation equation (Eq. 3).
>
> In addition, to address the protocol justification specifically raised here, we have added a new statement (in orange) clarifying that CAVS directly generalizes the well-established class-incremental segmentation (CIS) paradigm to the audio-visual domain. The disjoint/overlapped settings and background labeling strategy follow standard CIS conventions (MiB, PLOP), ensuring the protocol is not benchmark-specific but reflects real-world deployment scenarios.
>
> ---
>
> **Q2: Strengthen the analysis of CSR**
>
> **A2:** The sigmoid explanation has already been revised in response to Reviewer cyxw (after Eq. 12), clarifying its compressive role and why collision frequency is a principled replay criterion.
>
> To further address the concerns raised here, we have added the following:
>
> - **Threshold is not heuristic (in orange):** We have added an explicit clarification that the threshold $\mathcal{T}$ is defined as the *mean collision ratio across all learned classes*, it is not a manually tuned hyperparameter. This adaptive design means $\mathcal{T}$ automatically adjusts to the current task distribution. Since collision statistics are recomputed at each incremental stage, CSR naturally adapts to different task orderings without requiring threshold tuning.
>
> - **Why collision frequency works:** $\mathcal{F}_c$ directly measures how often old class $c$ is confused with new classes during inference. Classes with higher $\mathcal{F}_c$ have their decision boundaries actively disrupted by new class learning, making them most prone to catastrophic forgetting. Rehearsing them more frequently counteracts this boundary erosion.
>
> - **Sigmoid smoothing (already revised):** The sigmoid compresses the distribution to prevent a single highly-colliding class from dominating the resampling budget. After normalization, the result is a valid probability distribution. This is a standard smoothing technique, not a heuristic choice.
>
> ---
>
> **Q3: Improve fairness and clarity of experimental comparison**
>
> **A3:** We have already added detailed baseline adaptation documentation in Section 4.2, including: all baseline adaptation procedures (replacing visual-only backbone with our audio-visual encoder), controlled comparison variables (ResNet-50 backbone, $k=5$ per class, 30 epochs per task, batch size 2 per GPU on 4 NVIDIA L40 GPUs).
>
> Regarding disentangling the gain from replay design: the existing ablation (Tab. 3) already provides this decomposition, it compares no rehearsal (baselines), random rehearsal, MSS only, and MSS+CSR (full method), all under the same memory budget. The results clearly show: Random → MSS yields +2.0 mIoU, and MSS → MSS+CSR yields an additional +1.1 mIoU, demonstrating that the gains come specifically from our replay design rather than simply having more data.
>
> ---
>
> **Q4: Moderate or better support claims about modality entanglement**
>
> **A4:** We appreciate this concern. Our claims about modality entanglement are supported by multiple lines of existing evidence:
>
> 1. **Collision pair statistics** (Fig. A.4): The most frequently colliding class pairs (e.g., "guitar"–"man") correspond to frequently co-occurring objects, directly validating that co-occurrence causes cross-modal confusion.
> 2. **Ablation results** (Tab. 3): MSS (+2.0) and CSR (+1.1) improvements over random rehearsal demonstrate that modality-aware selection and collision-based resampling specifically target the entanglement problem.
> 3. **Qualitative results** (Fig. 7): Our method correctly segments previously learned classes where baselines exhibit confusion.
>
> Additionally, we have softened language where mechanistic evidence is indirect (in orange), e.g., "reduce forgetting on classes prone to modality entanglement" rather than claiming to directly "mitigate modality entanglement."
>
> ---
>
> **Q5: Improve writing and presentation**
>
> **A5:** The following revisions have already been made:
>
> 1. **Notation consistency:** A notation table (Tab. 1) defining all symbols before use.
> 2. **Notation fix:** Corrected $\mathcal{Y}$ to $\mathcal{Y}^t$ in Eq. 1.
> 3. **Citation fix:** Removed a dangling reference missing from the bibliography.
> 4. **Figure clarity:** Revised captions for Figures 3 and 4.
> 5. **Grammar:** Proofreading pass to fix grammatical errors and unclear sentences.
>
> ---

---

### Decision · Action_Editor_vvTy · 2026-06-05

**Recommendation:** Accept as is

**Additional Comments:**

The paper has improved substantially during revision. The authors have addressed the main reviewer concerns by clarifying the CAVS formulation, improving notation and presentation, documenting baseline adaptation and implementation details, adding analyses of modality entanglement and collision statistics, providing failure cases, discussing long-video limitations, and adding a comparison with an open-vocabulary AVS method.

The remaining limitations, including the heuristic nature of some design choices, the added complexity of rehearsal-based sample selection, and the limited exploration of long-video settings, are now sufficiently discussed and do not prevent acceptance under TMLR's criteria. The reviewers' final recommendations are consistently positive, and I therefore recommend acceptance.

**Audience:**

Yes

**Audience Explanation:**

The paper studies an interesting and timely problem at the intersection of continual learning, audio-visual learning, and semantic segmentation. Compared with existing multimodal continual learning settings that often focus on coarse-grained classification or related tasks, CAVS brings continual learning into a fine-grained pixel-level audio-visual segmentation scenario. This new task formulation is likely to be of interest to researchers working on continual learning, multimodal learning, audio-visual segmentation, and memory-based rehearsal strategies.

The proposed analysis of multi-modal semantic drift and co-occurrence confusion also provides useful insights for understanding why continual learning is challenging in audio-visual segmentation. Even if the methodological contribution is relatively incremental and rehearsal-based, the benchmark formulation, empirical findings, and targeted analyses should be valuable to at least part of the TMLR audience.

**Claims And Evidence:**

Yes

**Claims Explanation:**

The revised submission provides sufficient evidence to support its main claims. The paper introduces Continual Audio-Visual Segmentation (CAVS) as a fine-grained multimodal continual learning task and proposes the Collision-based Multi-modal Rehearsal (CMR) framework to address multi-modal semantic drift and co-occurrence confusion. The empirical evaluation covers multiple continual audio-visual segmentation scenarios and shows consistent improvements over adapted continual learning baselines. The ablation studies further support the contributions of the proposed Multi-modal Sample Selection (MSS) and Collision-based Sample Rehearsal (CSR) components.

The initial reviews raised concerns about task formulation, notation and presentation, fairness of baseline adaptation, the interpretation of modality entanglement, the role of rehearsal, possible visual bias in MSS, missing comparisons with open-vocabulary AVS methods, and limitations under long-video settings. The authors have addressed these concerns by clarifying the task protocol and label transformation, adding definitions and notation, improving implementation details, adding analysis of collision statistics and failure cases, including an open-vocabulary AVS comparison, and softening claims where mechanistic evidence is indirect. Overall, while some aspects of the method remain heuristic, the central claims are now supported by clear empirical evidence and adequate analysis.